# An Autonomous Power-Frequency Control Strategy Based on Load Virtual Synchronous Generator

**Jiangbei Han** [†] ⬤**, Zhijian Liu** [†]**, Ning Liang \*, Qi Song and Pengcheng Li**

Faculty of Electric Power Engineering, Kunming University of Science and Technology, Kunming 650500, China; jiangbeikust@163.com (J.H.); zhijiankust@163.com (Z.L.); 20030118@kust.edu.cn (Q.S.); lpclike@126.com (P.L.)
* Correspondence: ningwhu@kust.edu.cn
† These authors contributed to the work equally and should be regarded as co-first authors.

**Abstract:** With the increasing penetration of the hybrid AC/DC microgrid in power systems, an inertia decrease of the microgrid is caused. Many scholars have put forward the concept of a virtual synchronous generator, which enables the converters of the microgrid to possess the characteristics of a synchronous generator, thus providing inertia support for the microgrid. Nevertheless, the problems of active power oscillation and unbalance would be serious when multiple virtual synchronous generators (VSGs) operate in the microgrid. To conquer these problems, a VSG-based autonomous power-frequency control strategy is proposed, which not only independently allocates the power grid capacity according to the load capacity, but also effectively suppresses the active power oscillation. In addition, by establishing a dynamic small-signal model of the microgrid, the dynamic stability of the proposed control strategy in the microgrid is verified, and further reveals the leading role of the VSG and filter in the dynamic stability of microgrids. Finally, the feasibility and effectiveness of the proposed control strategy are validated by the simulation results.

**Keywords:** hybrid AC/DC microgrid; virtual synchronous generator; autonomous power-frequency control strategy; dynamic small-signal model

---

## 1. Introduction

In order to deal with the increasingly severe energy crisis, there is an urgent need to develop hybrid AC/DC microgrids to replace the traditional AC grids, thus improving the penetration of new energies [1,2]. However, as one of the key instruments in hybrid AC/DC microgrids, the low inertia characteristic of the converter greatly influences the power supply quality of AC loads [3,4]. In the traditional AC grids, the rotating mass of synchronous generators (SGs) can provide enough inertia for the system, and the damping characteristic will be supplied by the power loss and mechanical friction of the stator. Nevertheless, SGs have been replaced by the distributed generation units in hybrid AC/DC microgrids. Therefore, how to give the converter have the same characteristics as SGs is of great significance to the safe and stable operation of microgrids.

In recent years, many scholars have conducted comprehensive researches for improving the characteristics of grid-connected converters, thus enhancing the inertia and damping of the system. For instance, Prof. Zhong has put forward the concept of a virtual synchronous generator (VSG) [5–8]. In the context of the backbone power grid being the AC power grid, SGs are defined as the default mechanism of the system, so all players in this system have autonomous synchronous characteristics. Adding a second-order model of SGs into the control strategy of the power converter (hence the birth of the VSG) would be helpful in making power converters more efficient and friendlier to AC power grids. Based on Prof. Zhong's research, many ameliorative strategies have been proposed to enhance microgrid stability. In [9], a VSG-based auto disturbance rejection control strategy is proposed.

Simulation experiments verified the positive effects of the proposed control strategy in estimating and compensating the total disturbance of the system. Moreover, [10] presents an adaptive virtual inertia control strategy based on improved bang-bang control. On the one hand, it can make full use of the variability of the virtual inertia to reduce dynamic frequency deviation; on the other hand, the frequency stability of the system is improved by setting the steady-state frequency interval and the steady-state inertia. In [11], a self-adaptive inertia and damping combination control strategy is presented, which supports the short-term active power of the system by using the adaptive inertial damping, and realizes the adaptive inertial damping control of the microgrid primary frequency modulation. However, the secondary frequency adjustment capability of the VSG has not been fully exploited. At the same time, when the load power changes rapidly or multiple VSGs work simultaneously, there is a serious problem of system frequency deviation.

To enhance the power quality of the microgrid, many researchers have put forward a series of improved algorithms of VSGs to realize secondary frequency regulation. Briefly, [12] introduces a PI controller to the power-frequency control unit in the VSG so that the system can automatically realize secondary frequency regulation in island mode. In [13], by bringing two integrators in the virtual mechanical and damping torque part of the p-f controller of the traditional VSG, the system frequency can return to the rated value in an islanded microgrid. In addition, [14] proposes an integrated control algorithm, which combines a quasi-synchronization algorithm with an islanding detection algorithm, thus achieving frequent deviation-free regulation in both grid-connected and islanded microgrids.

From the studies mentioned above, it is not difficult to find that many VSG-based ameliorative control strategies have been proposed to realize secondary frequency regulation in microgrids. However, due to the rotor oscillation characteristics of VSGs, it is easy to cause the active power oscillation of the AC load, especially when multiple VSGs operate simultaneously. In response to this issue, a small-signal model of multiple VSG parallel operation is given in [15], which theoretically reveals the mechanism of power oscillation caused by the rotor motion equation. Moreover, the authors of [16] increase the damping ratio of the microgrid system by improving the rotor motion equation, thereby effectively suppressing power oscillations. However, the control coefficients need to be continuously modified during the control process, which further complicates the control system. Similarly, in [17], the power oscillation is suppressed by increasing the damping coefficient. Nevertheless, it will influence the stability of the system. In order to alleviate this issue, [18] effectively restrains the power oscillation by changing the inertia coefficient adaptively during the disturbance, thus improving the stability of the system. Additionally, in [19], the stability constraint of virtual inertia is derived, and the relationship between the ratio of damping and the stability is quantified. Moreover, the authors of [20] propose a polyhedral linear differential inclusion (PLDI) model, which is used to analyze the small signal stability of the microgrid under uncertain excitation. Although a series of methods have been proposed to improve the dynamic stability of the system, little research is conducted on the deeper relations among different subsystems in the microgrid. In addition, if multiple VSGs operate simultaneously, how to realize frequent deviation-free control while restraining active power oscillation also needs more concentration and research.

Motivated by the above studies, this paper mainly focuses on the possible modification of the microgrid-interface converter. A VSG-based autonomous power-frequency control strategy is proposed for the first time, which is applied to the hybrid AC/DC microgrid with multiple AC loads operate simultaneously. By improving the conventional VSG control strategy, the microgrid can not only realize frequent deviation-free control, but also effectively suppress the active power oscillation caused by different parameters of VSGs. Besides, a dynamic small-signal model of the microgrid with a single VSG is established, which further reveals the internal impact of each subsystem on the stability of the microgrid. To the best of the authors' knowledge, this paper makes the following contributions:

- The secondary frequency regulation controller is introduced into the VSG to realize frequent deviation-free control in microgrids.

- A proportional control is introduced into the output of the secondary frequency regulation controller, so as to allocate the system active power capacity according to the rated capacity of each VSG in microgrids. In addition, an active droop control is added to the input of the secondary frequency regulation controller, thus suppressing the active power oscillation caused by different operating parameters of each VSG.
- A dynamic small-signal model of the microgrid is established. Based on this model, the participation factors of the state variables in the dynamic model are calculated to reveal the internal relationship between the subsystems and the dynamic characteristic of microgrids.

The remaining of this paper is organized as follows. After the introduction, a VSG-based autonomous power-frequency control strategy is proposed in Section 2. Then, a dynamic small-signal model of the microgrid with a single VSG is established in Section 3. In Section 4, the dynamic stability of the model is analyzed, and then the simulation processes are given in Section 5. Finally, some brief conclusions are listed.

## 2. Autonomous Power-Frequency Control Strategy

To avoid the influence of the high-order synchronous generator model on the transient stability of the system, the classical second-order model of the synchronous generator is used to simulate the inertia and damping characteristics of the synchronous generator, and the rotor motion equation is expressed as

$$\begin{cases} J\frac{d\omega}{dt} = \frac{P_t - P_e}{\omega_N} - D\Delta\omega \\ \Delta\omega = \omega - \omega_N \\ \frac{d\theta}{dt} = \omega \end{cases} \tag{1}$$

where $\omega$ and $\omega_N$ are the actual and rated value of the rotor angular speed, respectively. $J$ represents the moment of inertia of the system. $\theta$ and $D$ represent the actual power angle of the system and the damping coefficient, respectively. Moreover, $P_t$ and $P_e$ represent the mechanical power and electromagnetic power of the VSG, respectively, and the electromagnetic power is defined as the instantaneous active power of the AC load. When the system is connected to the microgrid, the frequency of the AC load needs to be adjusted by the governor control, as manifested in Equation (2).

$$P_t = P_N + K_f(\omega_N - \omega) \tag{2}$$

where $P_N$ is the rated power of VSG, and $K_f$ is the coefficient of the governor control. The power-frequency control loop with primary frequency control (PFC) is shown in Figure 1.

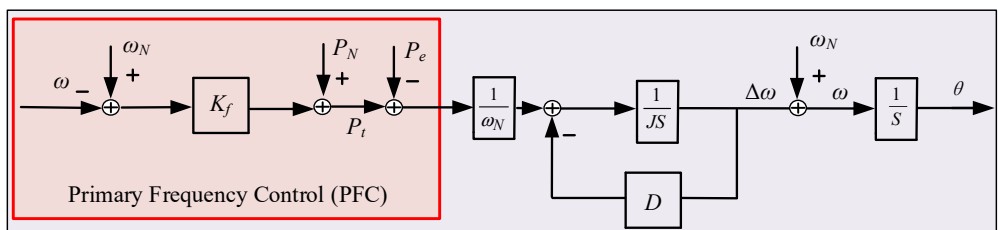

**Figure 1.** Primary frequency control.

Then, according to Equations (1) and (2), the transfer function in Figure 1 can be expressed as

$$\frac{\omega_N - \omega}{P_N - P_e} = -\frac{1}{J\omega_N S + D\omega_N + K_f} = -\frac{n}{TS + 1} = -\frac{1}{\frac{T}{n}S + \frac{1}{n}} \tag{3}$$

$$n = \frac{1}{D\omega_N + K_f}, T = J\omega_N n = \frac{J\omega_N}{D\omega_N + K_f} \tag{4}$$

where $n$ is the time constant, and $T$ is the droop coefficient. The transfer function is a first-order inertial link, which can simulate the inertial effect of the virtual synchronization algorithm and has the characteristics of primary frequency modulation. The power-frequency control loop with PFC would simulate the inertial response of the synchronous generator. Nevertheless, it has no capacity to realize the secondary frequency regulation under the large load fluctuation. Hence, in Figure 2, the secondary frequency regulation controller is introduced into the power-frequency control, which is used to replace the PFC. In view of the existence of the secondary frequency regulation controller, the power-frequency control loop can realize the frequent deviation-free control alone, so this control method is defined as the autonomous frequency control (AFC).

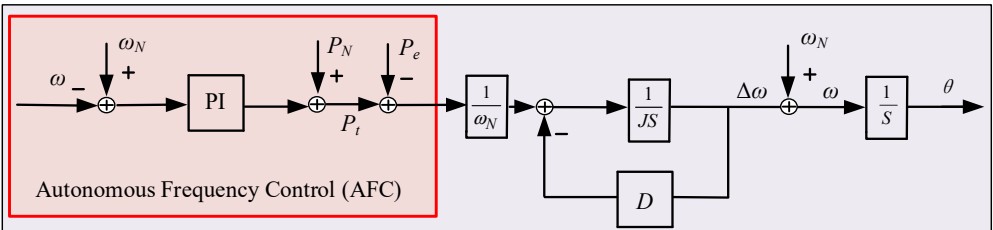

**Figure 2.** Autonomous frequency control.

The control equations of the power-frequency control with AFC are defined as

$$
\begin{cases}
J\frac{d\omega}{dt} = (P_t - P_e)\frac{1}{\omega_N} - D(\omega - \omega_N) \\
P_t = P_N + (K_f + \frac{K_i}{S})(\omega_N - \omega) \\
A = (\omega_N - \omega)\frac{K_i}{S}
\end{cases}
\tag{5}
$$

where $K_f$ and $K_i$ represent the proportional gain and integral gain of the secondary frequency regulation controller, and $A$ is the output of the integral loop. By changing Equation (5) and the Laplace transformation, Equations (6) and (7) can be obtained as

$$
\frac{\omega_N - \omega}{P_N - P_e} = \frac{1}{J\omega_N S^2 + (D\omega_N + K_f)S + K_i}
\tag{6}
$$

$$
\begin{cases}
\Delta P = [J\omega_N S^2 + (D\omega_N + K_f)S + K_i](\omega_N - \omega) \\
\Delta\omega = \frac{P_N - P_e}{J\omega_N S^2 + (D\omega_N + K_f)S + K_i}
\end{cases}
\tag{7}
$$

Then, according to the final theorem, Equation (8) is obtained as

$$
\begin{cases}
\lim_{t\to 0}\Delta P = \lim_{s\to\infty} S\Delta P = 0 \\
\lim_{t\to 0}\Delta\omega = \lim_{s\to\infty} S\Delta\omega = 0
\end{cases}
\tag{8}
$$

It can be found from Equation (8) that the AFC has the capacity to realize frequency secondary regulation when the microgrid operates with a single VSG. However, if multiple VSGs operate simultaneously in DC distribution grids (as shown in Figure 3), the corresponding communication coordination control should be taken for the VSG in the microgrid to ensure that each VSG can distribute the active power according to its rated value when participating in secondary frequency regulation. However, this method requires high reliability and accuracy of communication. Therefore, in Figure 4, a proportional control is added to the output of the secondary frequency controller to realize the optimal power distribution.

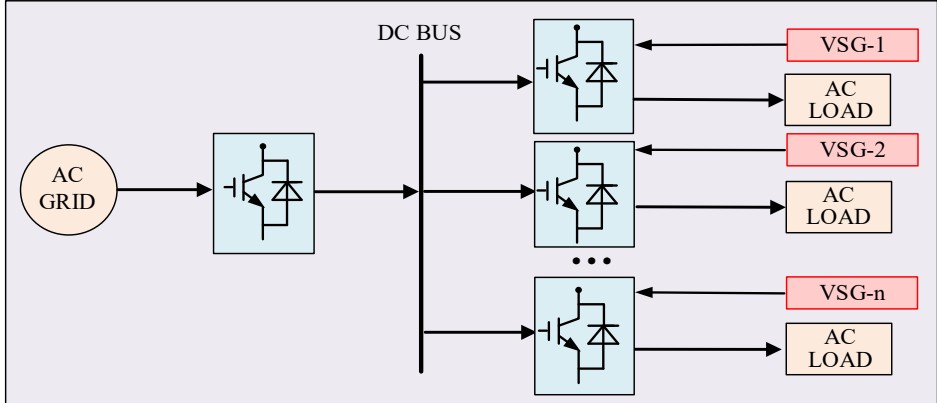

**Figure 3.** DC distribution grid.

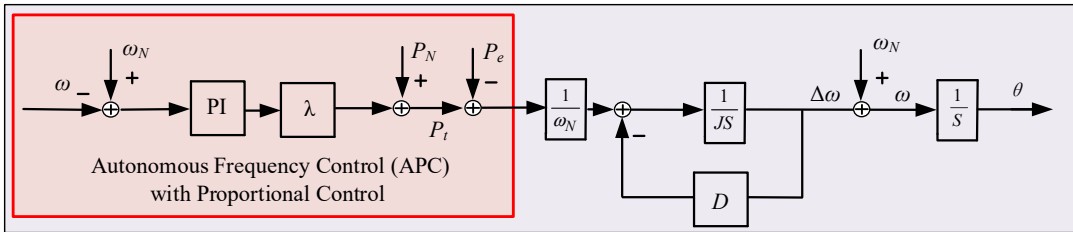

**Figure 4.** Autonomous frequency control with proportional control.

Supposing the rated power ratio of each VSG is defined as

$$P_{N1} : P_{N2} : \ldots : P_{Nn} = \delta_1 : \delta_2 \ldots : \delta_n \tag{9}$$

Then, the output power of VSG is

$$\begin{cases} P_t = P_N + \Delta P \\ \Delta P = (K_f + \frac{K_i}{s})(\omega_N - \omega) \end{cases} \tag{10}$$

Motivated by Equations (9) and (10), we can find that the active output power of VSG is not proportional to its rated capacity due to the secondary frequency regulation controller. Therefore, when the PI parameters of each VSG are the same, a proportional control is added to the output of the secondary frequency regulation controller in Figure 4. Moreover, the ratio of the coefficient of each proportional control λ is equal to the ratio of each VSG's rated capacity. That is

$$\lambda_1 : \lambda_2 : \ldots : \lambda_n = \delta_1 : \delta_2 : \ldots : \delta_n \tag{11}$$

Then, we can obtain the modified mathematical model of the secondary frequency regulation controller.

$$\Delta P = \lambda(K_f + \frac{K_i}{s})(\omega_N - \omega) \tag{12}$$

In addition, from Equations (9) to (12), it can be concluded that

$$P_{t1} : P_{t2} : \ldots : P_{tn} = \delta_1 : \delta_2 : \ldots : \delta_n \tag{13}$$

Hence, the above method can realize the optimal power allocation when multiple VSGs operate simultaneously. However, in view of the fact that the damping coefficient D of VSG is not the same, the dynamic response speed of each VSG is different. It is possible that VSGs with fast dynamic responses would occupy more active power capacity. Therefore, in this paper, based on the control strategy in

Figure 4, an active power droop control is added to the input of the secondary frequency control, as shown in Figure 5.

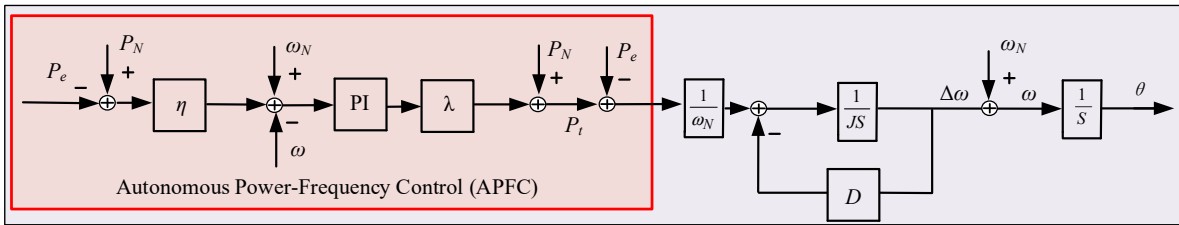

**Figure 5.** Autonomous power-frequency control (APFC).

Where $\eta$ is the coefficient of the active power droop control. In addition, the greater damping $D$ is, the faster the dynamic response speed the VSG possesses. Hence, according to the inverse ratio of the damping value in each VSG, the coefficient of the active power droop control $\eta$ is selected. In other words, the larger damping $D$ is, the smaller $\eta$ is, and vice versa. The mathematical model of the active power droop control can be expressed as

$$\begin{cases} \omega_{ref} = \omega_N + \eta(P_N - P_e) \\ \frac{\eta_1}{D_n} = \frac{\eta_2}{D_{n-1}} = \ldots = \frac{\eta_{n-1}}{D_2} = \frac{\eta_n}{D_1}(D_1 > D_2 > D_{n-1} > D_n) \end{cases} \tag{14}$$

By introducing active power droop control, the VSG with faster dynamic performance can possess the smaller active droop coefficient, and the larger active droop coefficient can be obtained by the VSG with slower dynamic performance. In conclusion, the autonomous power-frequency control (as shown in Figure 5) based on VSG (APFC-VSG) can restrain the active power oscillation caused by the dynamic response difference between VSGs. At the same time, the input power of each AC load can be distributed according to the rated capacity of each VSG, and the secondary frequency regulation would be realized with multiple AC loads operating simultaneously, so that the purpose of frequency regulation and active power distribution is achieved in microgrids.

## 3. Establishment and Analysis of Dynamic Small-Signal Model

To verify the dynamic stability of microgrids with APFC-VSG, a dynamic small-signal model is built, which includes the control circuit and electrical circuit. The control circuit consists of autonomous power-frequency control, excitation control, and virtual resistance. Moreover, the electrical circuit of microgrid includes a three-phase converter, LC-filter circuit, and AC load. Figure 6 shows the electrical circuit of microgrid and the control circuit of APFC-VSG. In this microgrid, the dynamic characteristics of the IGBT bridge and PLL are ignored, and the switch action is not considered. In addition, the output voltage of the bridge arm is equal to that of the controller. Based on the above model assumptions, the small-signal model of each part of the system is set up as follows.

*3.1. Autonomous Power-Frequency Control*

The following instantaneous active power $P$ and reactive power $Q$ can be obtained by transforming the three-phase voltage and current signals collected at the AC load to $d_{q0}$ coordinate system.

$$\begin{cases} P = \frac{3}{2}(u_d i_d - u_q i_q) \\ Q = -\frac{3}{2}(u_d i_q - u_q i_d) \end{cases} \tag{15}$$

By linearizing Equation (15), the small-signal model is obtained, as shown in Equation (16).

$$\begin{cases} \Delta P = [\frac{3}{2}i_d \frac{3}{2}i_q]_{1\times2}[\Delta u_{dq}]_{2\times1} + [\frac{3}{2}u_d \frac{3}{2}u_q]_{1\times2}[\Delta i_{dq}]_{2\times1} \\ \Delta Q = [-\frac{3}{2}i_q \frac{3}{2}i_d]_{1\times2}[\Delta u_{dq}]_{2\times1} + [\frac{3}{2}u_q \frac{3}{2}u_d]_{1\times2}[\Delta i_{dq}]_{2\times1} \end{cases} \tag{16}$$

Then, the autonomous power-frequency control in Figure 5 is used for frequency regulation. So, the following small-signal model is obtained.

$$\begin{cases} S[\Delta\omega] = -\Delta\theta\left(\frac{2\pi K_f}{J\omega_N} + \frac{D}{J}\right) - \frac{\Delta P}{J\omega_N} \\ S[\Delta A] = K_i\Delta\omega \end{cases} \quad (17)$$

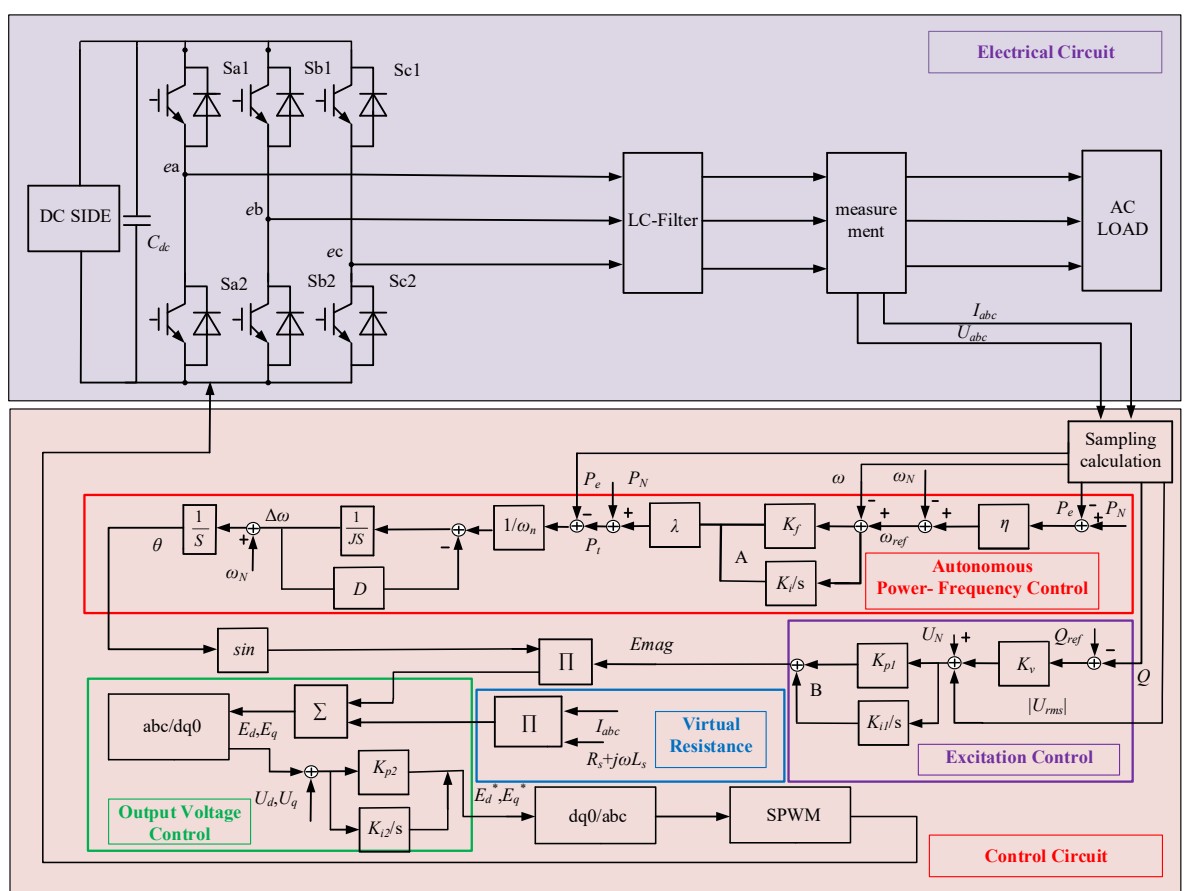

**Figure 6.** Microgrid with single virtual synchronous generator (VSG).

## 3.2. Excitation Control

The excitation control model consists of two parts: a reactive power droop model and an excitation voltage regulation model. The mathematical model can be expressed as

$$\begin{cases} U_{ref} = U_N + K_V(Q_{ref} - Q) \\ U_{rms} = \sqrt{u_d{}^2 + u_q{}^2} \\ E_{mag} = K_{p1}(U_{ref} - U_{rms}) + B \\ B = \frac{K_{i1}}{S}(U_{ref} - U_{rms}) \end{cases} \quad (18)$$

In this model, $U_{ref}$ represents the voltage reference value after reactive droop regulation, $U_N$ is the rated voltage value of the AC system, $K_V$ is the reactive droop regulation coefficient, $Q_{ref}$ and $Q$ represent the reactive power reference value and measured value respectively, $U_{rms}$ is the AC side voltage collected by the AC system which is acted as the excitation voltage involved in regulation, while $K_{P1}$ and $K_{i1}$ are the proportional and integral coefficients of the PI controller, respectively. The small-signal model of the excitation control is obtained as

$$\begin{cases} S[\Delta B] = -K_i K_V \Delta Q - K_i \Delta U \\ \Delta U = \left[ \dfrac{u_d}{\sqrt{u_d{}^2 + u_q{}^2}} \quad \dfrac{u_q}{\sqrt{u_d{}^2 + u_q{}^2}} \right]_{1\times 2} \end{cases} \tag{19}$$

### 3.3. Output Voltage Control

To ensure the stability of the final output voltage signal, the output voltage at the outlet side of VSG is taken as the reference value, and the voltage of the AC load is tracked by the PI controller. The control Equation (20) can be obtained.

$$\begin{cases} E_d{}^* = (E_d - u_d)K_{p2} + C \\ C = (E_d - u_d)\frac{K_{i2}}{S} \\ E_q{}^* = (E_q - u_q)K_{p2} + D \\ D = (E_q - u_q)\frac{K_{i2}}{S} \end{cases} \tag{20}$$

By linearizing Equation (20), the small-signal model is expressed as

$$\begin{cases} S[\Delta C] = K_{i2}\Delta E_d \\ S[\Delta D] = K_{i2}\Delta E_q \end{cases} \tag{21}$$

### 3.4. LC-Filter

According to Kirchhoff law and the electric circuit of the microgrid in Figure 6, the LC filter circuit model can be derived as

$$\begin{cases} L_r \frac{di_{Labc}}{dt} = U_{iabc} - U_{abc} - i_{Labc} R_r \\ C_r \frac{dU_{abc}}{dt} = i_{Labc} - i_{abc} \end{cases} \tag{22}$$

where $L_r$ is the filter inductance, $C_r$ is the filter capacitance, $U_{iabc}$ is the three-phase voltage of the input filter circuit, and $i_{Labc}$ is the three-phase current of the input filter circuit. The following equations can be obtained by projecting the three-phase sinusoidal quantities into the rotating coordinate system.

$$\begin{cases} L_r \frac{di_{Ld}}{dt} = U_{id} - u_d - i_{Ld}R_r + i_{Lq}\omega_N L_r \\ L_r \frac{di_{Lq}}{dt} = U_{iq} - u_q - i_{Lq}R_r + i_{Ld}\omega_N L_r \\ C_r \frac{du_d}{dt} = i_{Ld} - i_d + u_q \omega_N C_r \\ C_r \frac{du_q}{dt} = i_{Lq} - i_q + u_d \omega_N C_r \end{cases} \tag{23}$$

The linearized small signal model is expressed as

$$s\left[\Delta i_{\text{Ldq}}\right]_{2\times 1} = \begin{bmatrix} -\frac{R_r}{L_r} & \omega_g \\ -\omega_g & -\frac{R_r}{L_r} \end{bmatrix}_{2\times 2}\left[\Delta i_{\text{Ldq}}\right]_{2\times 1} + \begin{bmatrix} \frac{1}{L_r} & 0 \\ 0 & \frac{1}{L_r} \end{bmatrix}_{2\times 2}\left[\Delta u_{\text{idq}}\right]_{2\times 1} - \begin{bmatrix} \frac{1}{L_r} & 0 \\ 0 & \frac{1}{L_r} \end{bmatrix}_{2\times 2}\left[\Delta u_{\text{dq}}\right]_{2\times 1} \tag{24}$$

$$s\left[\Delta u_{\text{dq}}\right]_{2\times 1} = \begin{bmatrix} \frac{1}{C_r} & 0 \\ 0 & \frac{1}{C_r} \end{bmatrix}_{2\times 2}\left[\Delta i_{\text{Ldq}}\right]_{2\times 1} - \begin{bmatrix} \frac{1}{C_r} & 0 \\ 0 & \frac{1}{C_r} \end{bmatrix}_{2\times 2}\left[\Delta i_{\text{dq}}\right]_{2\times 1} + \begin{bmatrix} \omega - \omega_N & 0 \\ 0 & \omega_N - \omega \end{bmatrix}_{2\times 2}\left[\Delta u_{\text{dq}}\right]_{2\times 1} \tag{25}$$

### 3.5. AC Load

According to Kirchhoff law, the following equation is derived.

$$i_{md} = \frac{u_d + \omega L_m i_{mq}}{L_m s + R_m} \tag{26}$$

$$i_{mq} = \frac{u_q - \omega L_m i_{md}}{L_m s + R_m} \tag{27}$$

where $L_m$ is the load inductance, and $R_m$ is the load resistance. The small-signal model of the AC load is expressed as

$$s\left[\Delta i_{mdq}\right]_{2\times 1} = \begin{bmatrix} -\frac{R_m}{L_m} & \omega \\ -\omega & -\frac{R_m}{L_m} \end{bmatrix}_{2\times 2} \left[\Delta i_{mdq}\right]_{2\times 1} + \begin{bmatrix} \frac{1}{L_m} & 0 \\ 0 & \frac{1}{L_m} \end{bmatrix} \left[\Delta u_{dq}\right]_{2\times 1} \quad (28)$$

Based on the above model, the microgrid with a single VSG is shown in Figure 6. The DC side is a 750 V medium voltage DC bus, and the AC side is a 220 V, 50 Hz system. The AC load active power is 10 kW, and the reactive power is 100 var. The AC system is connected to the DC distribution system through the converter, and the system parameters are shown in Table A1. The model consists of 11 state variables: $\omega$, A, B, C, D, $i_{Ld}$, $i_{Lq}$, $u_d$, $u_q$, $i_{md}$, $i_{mq}$. The above state variables represent the dynamic characteristics of the system and cooperate with the algebraic equations to form the full order dynamic model of microgrids.

According to the dynamic model of microgrids, Lyapunov's stability theory is used to analyze the stability of the microgrid. Firstly, the stable operation point of the system is obtained in Table 1. Secondly, the dynamic model is linearized at the system balance point. The stable operating point is brought into the system state space equations, and the Jacobian matrix of the system state space equations is obtained. The eigenvalues of the Jacobian matrix calculated by MATLAB are in Table 2. The distribution of eigenvalues is shown in Figure 7.

**Table 1.** Stable operation points of the systems.

| State Variable | Stable Operating Point | State Variable | Stable Operating Point |
|---|---|---|---|
| $\omega$(rad/s) | 314.13 | D | 0.023 |
| $u_d$(V) | 323.27 | $i_{Ld}$ (A) | 18.06 |
| $u_q$ (V) | −22.32 | $i_{Lq}$ (A) | −22.99 |
| A | 3.12 | $i_{md}$ (A) | 15.43 |
| B | 0.09 | $i_{mq}$ (A) | −17.89 |
| C | 0.004 | | |

**Table 2.** Eigenvalues of the microgrid.

| Number | Characteristic Value |
|---|---|
| 1 | −0.671 |
| 2 | −0.0001 |
| 3,4 | −998.5888 ± 313.9922$i$ |
| 5,6 | −272.4938 ± 15,269.7186$i$ |
| 7,8 | −272.4852 ± 14,641.6651$i$ |
| 9 | −992.4683 |
| 10,11 | −0.00113 ± 6.8096$i$ |

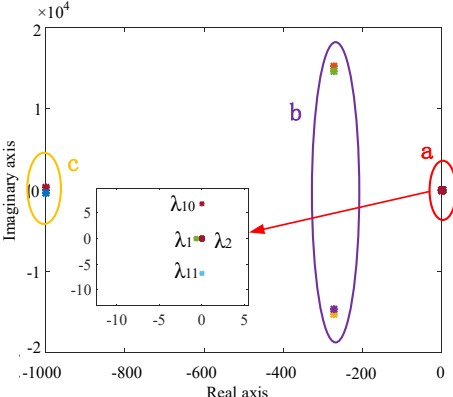

**Figure 7.** Distribution of the eigenvalues.

As shown in Figure 7, the characteristic root of region *a* is the closest to the virtual axis, so it has a great influence on the system stability. Region *b* exists in the form of conjugates, and regions *b* and

*c* are far away from the virtual axis, so the eigenvalues of regions *b* and *c* have little influence on the stability of the system. Each eigenvalue in the Jacobian matrix has corresponding left eigenvector $\alpha$ and right eigenvector $\beta$. To explain the correlation between state variables and the eigenvalues in region *a*, the participation factors corresponding to each state variable can be obtained by calculating the participation matrix corresponding to the Jacobian matrix. $P_{ij}$ is the participation factor. As $P_{ij}$ gradually increases, it indicates that the influence of the state variable *j* on the response mode *i* corresponding to the characteristic value gradually increases. By judging the influence degree of different state variables on eigenvalues, we can find the influence of each sub-model on the stability of the system. In this case, it is considered that the participation factor greater than 0.001 has a secondary impact on system stability, and greater than 0.01 has a primary impact. The expression of participation factor after normalized eigenvalue is as follows.

$$P_{ij} = \frac{\left|\alpha^{T}ij\right|\left|\beta_{ij}\right|}{\sum_{1}^{N}\left(\left|\alpha^{T}kj\right|\left|\beta_{kj}\right|\right)} \tag{29}$$

where *j* and *i* represent the state variable and the response mode, respectively, and *N* is the number of the eigenvalues. In addition, $\alpha_{kj}$ represents the left eigenvector of the *j*-th state variable in the *k*-th mode, and $\beta_{kj}$ represents the right eigenvector of the *j*-th state variable in the *k*-th mode. Through Equation (29), the participation factors of each state variable corresponding to the eigenvalue of area *a* can be calculated. From the distribution of the eigenvalues in Figure 7, the eigenvalue of area *a* is 1, 2, 10, and 11, respectively. The participation factors of each state variable corresponding to the dominant eigenvalue are shown in Table 3.

**Table 3.** Participation factors of the dominant eigenvalues.

| State Variable | Participation Factors of Eigenvalues 1 and 2 | Participation Factors of Eigenvalues 10 and 11 |
|:---:|:---:|:---:|
| $\omega$ | 0.5675 | 0.0327 |
| A | 0.3127 | 0.3241 |
| B | 0.0016 | 0.1463 |
| $i_{Ld}$ | 0.0013 | 0.2536 |
| $i_{Lq}$ | 0.0025 | 0.1396 |

It can be seen from Table 3 that the dominant eigenvalues 1 and 2 are mainly influenced by state variables A and $\omega$, which are in the autonomous power-frequency control. The dominant eigenvalues 10 and 11 are mainly affected by state variables A in the autonomous power-frequency control model, B in the excitation control model, $i_{Ld}$ and $i_{Lq}$ in the filter. Therefore, when there is a big disturbance in the system, the system can run stably by adjusting the relevant parameters of the autonomous power-frequency control, the excitation control model, and the LC filter.

## 4. Simulation Results

### 4.1. The Microgrid with Single VSG

To verify the regulation performance of primary frequency control (PFC) and autonomous frequency control (AFC) in the microgrid with a single converter, the microgrid model with a single converter is established in MATLAB/Simulink environment. In this model, the DC side is a 750 V medium voltage DC bus, the AC side is a 220 V/50 Hz system, the AC load active power and reactive power are 10 kW and 100 var, respectively; other simulation parameters are shown in Table A1. PFC and AFC are used in the power frequency control of the VSG algorithm to form two converter control strategies, respectively, hence the birth of the PFC-VSG and AFC-VSG. The regulation performance of VSG is simulated and analyzed under the sudden AC load increase and the sudden AC load decrease.

### 4.1.1. Sudden Load Increase Condition

When the active power load increases from 10 to 12 kW in 0.2 s, the active power waveform and frequency waveform of the two control strategies are observed, as shown in Figures 8 and 9. As shown in Figure 8, when the active power load suddenly increases by 2 kW, PFC-VSG cannot provide enough active power to the AC system, so the AC system can only operate at the active power of 10.05 kW. AFC-VSG can provide enough active power for the load. Because of the PI controller in AFC-VSG, the mechanical power provided by the virtual synchronous generator can track the fluctuation of load power rapidly, which makes the active power rapidly rise from 10 to 12 kW. As shown in Figure 9, PFC-VSG reflects the inertia characteristics of VSG. When the active load suddenly increases, the frequency of corresponding VSG decreases, meeting the inertia characteristics of a synchronous generator. Nevertheless, the frequency deviation could even reach −0.04 Hz, and it cannot return to the initial operating point to continue operation, which will cause damage to the equipment with higher power quality requirements. Compared with PFC-VSG, AFC-VSG has the same inertia characteristics, but its frequency deviation is smaller. Meanwhile, AFC-VSG, which has excellent adjustment characteristics, can return the frequency to the rated value in a short time.

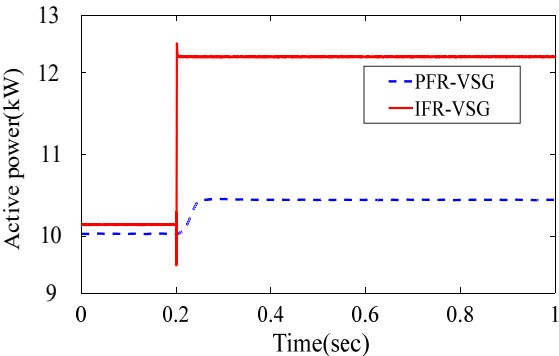

**Figure 8.** Active power regulation ability between the power-frequency control (PFC)-VSG and the autonomous frequency control (AFC)-VSG under the sudden load increase condition.

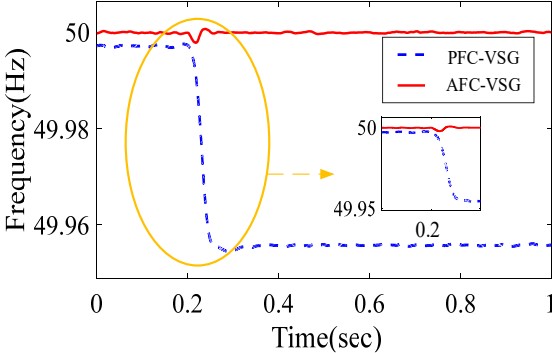

**Figure 9.** Frequency regulation ability between the PFC-VSG and the AFC-VSG under the sudden load increase condition.

### 4.1.2. Sudden Load Decrease Condition

When the active load decreases from 10 to 8 kW in 0.2 s, the active power and frequency waveforms of the two control strategies are observed, as shown in Figures 10 and 11. As shown in Figure 10, when the active load is suddenly reduced by 2 kW, PFC-VSG provides the AC system with excess active power, so that the AC system can only operate at the active power of 9.5 kW. Due to the PI controller in AFC-VSG, the mechanical power provided by the virtual synchronous generator rapidly tracks the fluctuation of the load power, which makes the active power rapidly reduce from 10 to 8 kW. As shown in Figure 11, PFC-VSG reflects the inertia characteristics of VSG. When the active load

drops suddenly, the frequency of corresponding VSG increases, meeting the inertia characteristics of a synchronous generator. Nevertheless, the frequency deviation could even reach 0.04 Hz, and it cannot return to the initial operating point to continue operation, which will cause damage to the equipment with higher power quality requirements. Compared with PFC-VSG, AFC-VSG has the same inertia characteristics, and its frequency deviation is smaller. Meanwhile, AFC-VSG, which has excellent adjustment characteristics, can return the frequency to the rated value in a short time.

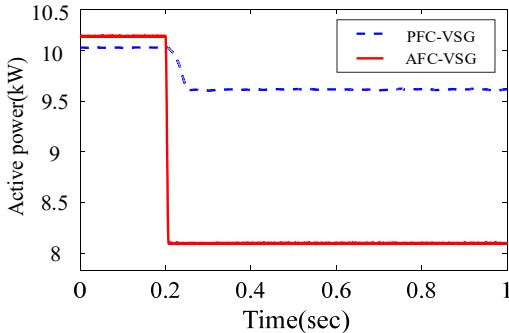

**Figure 10.** Active power regulation ability between the PFC-VSG and the AFC-VSG under the sudden load decrease condition.

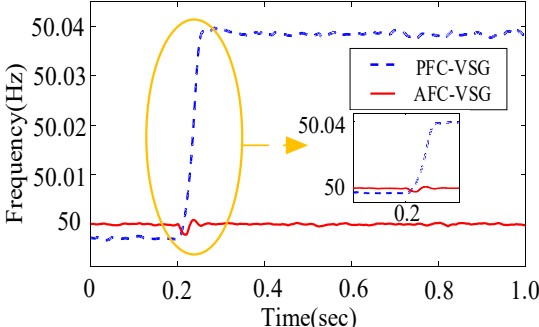

**Figure 11.** Frequency regulation ability between the PFC-VSG and the AFC-VSG under the sudden load decrease condition.

### 4.2. The Microgrid with Multiple VSGs

To verify the regulation performance among the PFC-VSG, AFC-VSG, and APFC-VSG in the microgrid with multiple converters, the microgrid model with three converters is established in MATLAB/Simulink environment (as shown in Figure 3), and part of simulation parameters are shown in Table 4.

**Table 4.** Part of simulation parameters.

| VSG Number | Rated Value of The Active Power (kW) | Damping Coefficient $D$ (W/rad$^2$) | Active Power Droop Control Coefficient $\eta$ | The Coefficient of Proportional Control $\lambda$ |
|---|---|---|---|---|
| VSG-1 | 10 | 20 | 500 | 1 |
| VSG-2 | 6 | 15 | 750 | 0.6 |
| VSG-3 | 10 | 10 | 1000 | 1 |

From the analysis results for the AFC-VSG in Section 2, we can find that the active output power of VSG is not proportional to its rated capacity due to the secondary frequency regulation controller. In order to realize the optimal power allocation when multiple VSGs operate simultaneously, we set the active power droop control coefficient $\eta$ according to the inverse ratio of the VSG damping coefficient. Moreover, the ratio of the coefficient of each proportional control $\lambda$ is equal to the ratio of each VSG's

rated value in Table 4. Then, a load fluctuation is added to each VSG when the simulation time is 0.1 s. The active power of VSG-1 decrease from 10 to 8 kW. The active power of VSG-2 rises from 6 to 7 kW, and the active power of VSG-3 rises from 10 to 11 kW. The comparison of the frequency regulation performance of each control strategy is given in Figure 12.

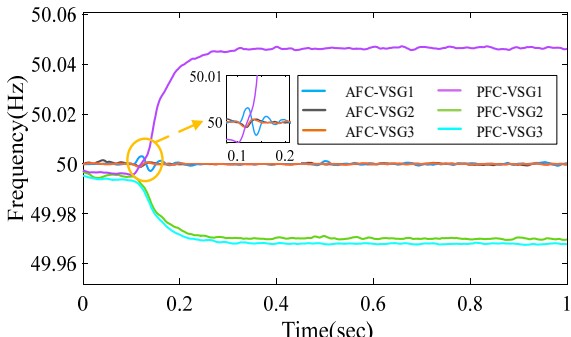

**Figure 12.** Frequency regulation ability between the PFC-VSG and the AFC-VSG.

Compared to the PFC-VSG control strategy, by adding proportional control to the output of the AFC-VSG, the microgrid can realize secondary frequency control when three VSGs operate simultaneously, and the frequency deviation is very small compared with the PFC-VSG. At the same time, in Figure 13, the proportional control in VSG allocates the system's active power capacity according to the rated capacity of each VSG in the microgrid. However, as shown in Figure 13, the active power oscillates, which is caused by the different damping provided by each VSG. Hence, the APFC-VSG proposed in Section 2 is used to replace the AFC-VSG. The APFC-VSG is constructed by adding the active droop controller to the input of the secondary frequency control controller. In order to eliminate the dynamic difference among multiple VSGs, which is caused by the damping coefficient, the coefficient $\eta$ of the active power droop controller is selected according to the inverse ratio of the damping $D$ value in each VSG.

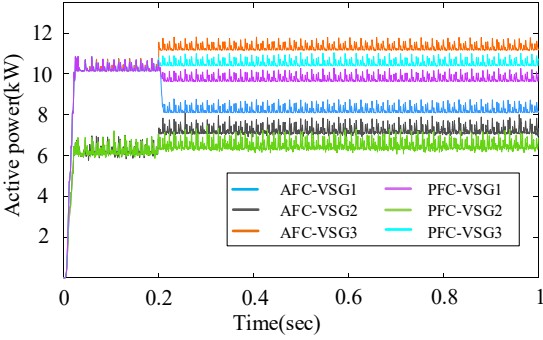

**Figure 13.** Active power regulation ability between the PFC-VSG and the AFC-VSG.

In Figure 14, the APFC-VSG has the capacity to solve the active power oscillation when multiple VSGs operate simultaneously. This is because the active droop controller is added to the input of the secondary frequency control controller, and the droop coefficient is set according to the inverse ratio of the VSG damping coefficient, so that there is no dynamic response difference between the VSGs, and all of them would provide sufficient inertia according to the rated capacity of AC load. The proposed APFC-VSG not only realizes the secondary frequency regulation but also has the ability to suppress oscillations, which provides double insurance for the safe and stable operation of the microgrid.

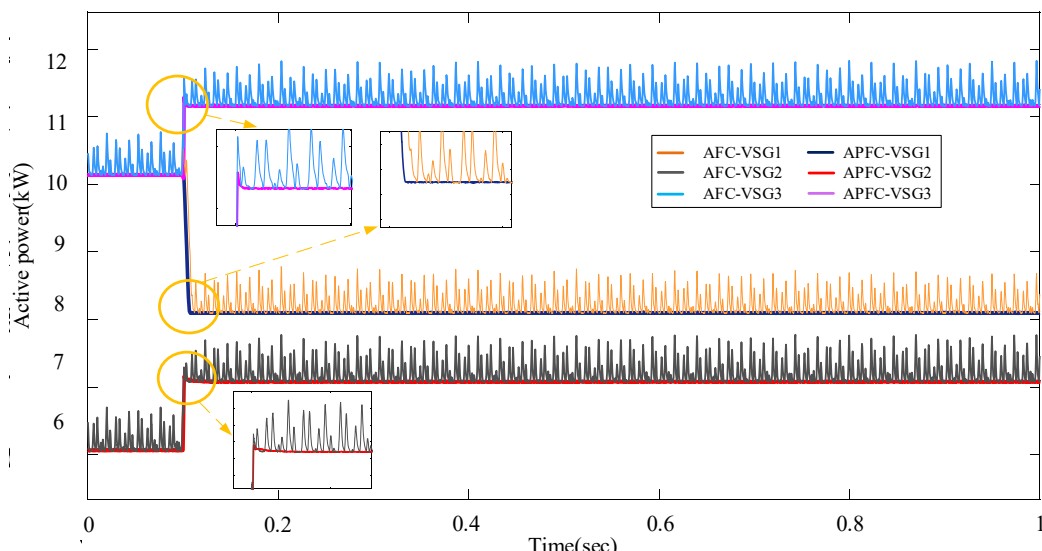

**Figure 14.** Active power regulation ability between the AFC-VSG and the APFC-VSG.

## 5. Conclusions

This paper proposes a modified APFC-VSG control strategy, which can not only realize the frequency secondary regulation of microgrids, but also effectively restrain the active power oscillation when multiple VSGs operate simultaneously. Moreover, by establishing the dynamical small-signal model of the microgrid with a single APFC-VSG, the dynamical stability of the microgrid is verified. Meanwhile, the leading roles of the APFC-VSG and filter in the dynamic stability of microgrid are further revealed. The detailed conclusions are as follows:

- Due to the existence of the secondary frequency regulation controller, the APFC-VSG can realize secondary frequency regulation when the microgrid operates with a single VSG.
- This paper adds the proportional control and active power droop control to the input and output of the secondary frequency regulation controller, respectively, which make the APFC-VSG not only have the capacity of achieving secondary frequency regulation but also to suppress the active power oscillation when multiple VSGs operate simultaneously in the microgrid.
- By establishing the dynamic small-signal model of the microgrid, the participation factors of the state variables in the dynamic model are calculated, which reveal the leading roles of the APFC-VSG and filter in the dynamic stability of the microgrid.
- The simulation results indicate that the APFC-VSG would provide enough inertia according to the rated capacity of each AC load in the microgrid. In particular, the NARC-VSG can still suppress the active power oscillation caused by the different damping coefficients when multiple VSGs operate simultaneously.

Possible future works include that the idea of the proposed control strategy should be further investigated to enhance the ability of reactive power regulation in microgrids.

**Author Contributions:** Author Contributions: Methodology, J.H. and Z.L.; Software, Q.Y.; Formal Analysis, Q.S.; Investigation, J.H. and P.L.; Resources, J.H.; Data Curation, P.L.; Writing-Original Draft Preparation, Z.L. and J.H.; Project Administration, J.H. and Z.L.; Funding Acquisition, Z.L. and N.L. All authors have read and agreed to the published version of the manuscript.

**Funding:** This work was supported by the Yunnan Human Resources Training Foundation of China (No. KKSY201904013).

**Conflicts of Interest:** The authors declare no conflict of interest.

## Appendix A

**Table A1.** Parameter of the System.

| System Parameter | Parameter Value | System Parameter | Parameter Value |
|---|---|---|---|
| $\omega_N$ (rad/s) | 314 | $K_{i2}$ | 50 |
| $P_N$ (kW) | 10 | $D$ (W/rad$^2$) | 20 |
| $Q_{ref}$ (var) | 100 | $J$ (kg/m$^2$) | 0.5 |
| $K_f$ | 1000 | $R_s/\Omega$ | 0.01 |
| $K_i$ | 0.01 | $L_s$/H | 0.0002 |
| $U_N$ (kV) | 0.38 | C/F | 0.00001 |
| $K_{p1}$ | 0.5 | $L_r$/H | 0.01838 |
| $K_{i1}$ | 5000 | $R_m/\Omega$ | 144.4 |
| $K_v$ (var/V) | 0.01 | $L_m$/H | 0.1444 |
| $K_{p2}$ | 400 | | |

All controller gains in Table A1 are designed as follows

1. $K_f$

The proportional coefficient $K_f$ increase from 0 until the system oscillates. Then, from the $K_f$ at this time gradually decreases until the system oscillation disappears, record the value of $K_f$ when the system oscillation disappears, and set the proportional coefficient $K_f$ of the proportional controller to 60%–70% of the $K_f$ Value.

2. $K_{p1}$, $K_{p2}$

When the parameter $K_p$ is determined in the PI controller, the integral term is first removed to make it a simple proportional regulation. The input is set to 60%–70% of the maximum allowable output value, and the proportional coefficient $K_p$ increases from 0 until the system oscillates. Then, from $K_p$ at this time gradually decreases until the system oscillation disappears, record the value of $K_p$ when the system oscillation disappears, and set the proportional coefficient of the PI controller to 60%–70% of the $K_p$ Value.

3. $K_i$, $K_{i1}$, $K_{i2}$.

After the proportional coefficient $K_p$ is determined, set a large integral constant $T_i$, in which $K_p/T_i$ = $K_i$, then gradually reduce $T_i$ until the system oscillates, and then gradually increase $T_i$ in turn until the system reached its stable operating point again, record the $T_i$ value when the oscillation disappears, and set $T_i$ of the PI controller to 150%–180% of this $T_i$ value, thus obtaining the $K_i$.

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
