# Peer review of "An Autonomous Power-Frequency Control Strategy Based on Load Virtual Synchronous Generator"

_processes, doi:10.3390/pr8040433_

Round 1
Reviewer 1 Report
Authors should improve the state of art. There are more related works.
Missing details how theoretical showed framework was implemented in the simulation
Table A1 is not referenced in the text
Author Response
Dear Editors and Reviewers:
Thank you for your letter and for the reviews’ comments concerning our manuscript entitled “An Autonomous Power-Frequency Control Strategy Based on Load Virtual Synchronous Generator” (ID: processes-758919). Those comments are all valuable and helpful for revising and improving our paper, as well as the important guiding significance to our researches. We have studied comments carefully and have made correction which we hope meet with approval. Revised portions are marked in red in the paper. The corrections in the paper and the responds to the reviewer’s comments are as follows.
Point 1: Authors should improve the state of art. There are more related works.
Responses: Thank you for your instructive suggestions. After carefully reading the introduction, we find that the references cited before are not enough related to the relevance of this paper. All the studies in this paper are based on VSG technology. Therefore, the concept of VSG should be introduced detailed, and the control method of micro-grid converter which is not in line with the technical route of this paper should not be introduced. In addition, we found that the previous cited articles are not the latest research results. Therefore, we cited some of the latest research results that are more consistent with the technical route as references. What’s more, this paper is dedicated to improving the active power oscillation caused by multiple VSG parallel connection based on realizing secondary frequency modulation of the micro-grid. The active power oscillation of multiple VSG grid connected is caused by different dynamic characteristics of each VSG.
1.Before modification
In recent years, many scholars have conducted comprehensive researches for improving the characteristics of grid-connected converters, thus enhancing the inertia and damping of the system. The existing modified control strategies of the grid-connected converters mainly include droop control and VSG control strategy [7]. However, in the islanded microgrid, it is difficult for the droop-based converters to provide inertia for the microgrid, which is more sensitive to the load fluctuation and makes the frequency change dramatically [8]. In contrast, the VSG-based converters have the capacity of changing the frequency smoothly in the same situation. Therefore, the VSG control strategy can satisfy the dynamic characteristics of the islanded microgrids. Moreover, VSG technology is mainly composed of the voltage-controlled VSG and current-controlled VSG [9]. In the grid-connected microgrid, both two strategies can effectively restrain the load fluctuations in the power grid. However, in the islanded microgrid, a current-controlled VSG only equates VSG with a current source, which cannot provide complete voltage and frequency support for the microgrid. On the contrary, the voltage-controlled VSG is designed according to the rotor motion equation and reactive voltage regulation control characteristics, which can better simulate the external characteristics of synchronous generators [10].
After modification
Page 1 Line 35-54
In recent years, many scholars have conducted comprehensive researches for improving the characteristics of grid-connected converters, thus enhancing the inertia and damping of the system. For instance, Prof. Zhong puts forward the concept of VSG [5-8]. In the context of backbone power grid being the AC power grid, SGs are defined as the default mechanism of the system, so all players in this system have autonomous synchronous characteristics. Adding a second-order model of SGs into the control strategy of the power converter, hence the birth of the virtual synchronous generator (VSG), would be helpful in making power converters more efficient and friendlier to AC power grid. Based on Prof. Zhong's researches,many ameliorative strategies have been proposed to enhance the microgrid stability. In [9], a VSG-based auto disturbance rejection control strategy is proposed. Simulation experiments verify the positive effects of the proposed control strategy in estimating and compensating the total disturbance of the system. Moreover, reference [10] presents an adaptive virtual inertia control strategy based on improved bang-bang control. On the one hand, it can make full use of the variability of the virtual inertia to reduce dynamic frequency deviation. On the other hand, the frequency stability of the system is improved by setting the steady-state frequency interval and the steady-state inertia. In [11], a self-adaptive inertia and damping combination control strategy is presented, which supports the short-term active power of the system by using the adaptive inertial damping, and realizes the adaptive inertial damping control of the microgrid primary frequency modulation. However, the secondary frequency adjustment capability of the VSG has not been fully exploited. At the same time, when the load power changes rapidly or multiple VSGs work simultaneously, there will be a serious problem of system frequency deviation.
- Zhong Q. C. Power Electronics Enabled Autonomous Power Systems: Architecture and Technical Routes. IEEE Trans. Ind. Electron. 2017, 64, 5907–5918.
- Zhong Q. C; Weiss G. Synchronverters: Inverters That Mimic Synchronous Generators. IEEE Trans. Ind. Electron. 2011, 58, 1259-1267.
- Zhong Q. C. Virtual Synchronous Generators and Autonomous Power Systems. Proc. CSEE 2017, 37, 336-348.
- Lv Z.P.; Sheng W. X.; Zhong Q. C.; Liu H. T.; Zeng Z.; Yang L.; Liu L. Virtual Synchronous Generator and Its Applications in Microgrid. Proc. CSEE 2014, 16, 2591-2603.
- Yu Y.; Hu X. Active Disturbance Rejection Control Strategy for Grid-Connected Photovoltaic Inverter Based on Virtual Synchronous Generator. IEEE Access 2019, 7, 17328-17336.
- Li J.; Wen B. Y.; Wang H. Y. Adaptive Virtual Inertia Control Strategy of VSG for Micro-Grid Based on Improved Bang-Bang Control Strategy. IEEE Access 2019, 7, 39509-39514.
- Li D. D.; Zhu Q. W.; Lin S. F.; Bian X. Y. A Self-Adaptive Inertia and Damping Combination Control of VSG to Support Frequency Stability. IEEE Trans Energy Convers 2017, 32, 397-398.
2.Before modification
In fact, many ameliorative strategies of the voltage-controlled VSG have been proposed to enhance the microgrid stability. In [11], a VSG-based auto disturbance rejection control strategy is proposed. Simulation experiments verify the positive effects of the proposed control strategy in estimating and compensating the total disturbance of the system. Moreover, reference [12] presents an adaptive virtual inertia control strategy based on improved bang-bang control. On the one hand, it can make full use of the variability of the virtual inertia to reduce dynamic frequency deviation. On the other hand, the frequency stability of the system is improved by setting the steady-state frequency interval and the steady-state inertia. References [11,12] are devoted to the adaptive adjustment of inertia, but the influence of damping factor on the stability of the system is not considered. To conquer this issue, a self-adaptive inertia and damping combination control strategy is presented in [13], which supports the short-term active power of the system by using the adaptive inertial damping, improves the frequency stability of the system by using the interleaving control technology, fills the blank in [11,12], and realizes the adaptive inertial damping control of the microgrid primary frequency modulation. However, the secondary frequency adjustment capability of the VSG has not been fully exploited. At the same time, when the load power changes rapidly or multiple VSGs work simultaneously, there will be a serious problem of system frequency deviation.
After modification
Page 2 Line 55-63
To enhance the power quality of the microgrid, many researchers put forward a series of improved algorithm of VSG to realize the secondary frequency regulation. Reference [12] introduces a PI controller to the power-frequency control unit in the VSG, so that the system can automatically realize the secondary frequency regulation in island mode. In [13], by bringing two integrators in the virtual mechanical and damping torque part in the p-f controller of the traditional VSG, the system frequency can return to the rated value in an islanded microgrid. In addition, reference [14] proposes an integrated control algorithm, which combines a quasi-synchronization algorithm with an islanding detection algorithm, thus achieving the frequent deviation-free regulation in both grid-connected microgrid and islanded microgrid.
- Xu Y.; Zhang T. F.; Yue D. Frequent Deviation-Free Control for Micro-Grid Operation Modes Switching Based on Virtual Synchronous Generator. LSMS/ICSEE, Nanjing, Jiangsu, China, 22-24 Sept. 2017.
- Jiang K.; Su H. S.; Sun D. Secondary Frequency Regulation Scheme Based on Improved Virtual Synchronous Generator in an Islanded Microgrid. Journal of Vibroengineering 2019, 21, 215-227.
- Shi K.; Zhou G. L.; Xu P. F.; Ye H. H.; Tan F. The Integrated Switching Control Strategy for Grid-Connected and Islanding Operation of Micro-Grid Inverters Based on a Virtual Synchronous Generator. Energies 2018, 11, 1544.
Point 2: Missing details how theoretical showed framework was implemented in the simulation
Responses: Thank you for your instructive suggestions. according to your kind suggestion, we have added the details in the simulation as follows.
1.Before modification
To verify the regulation performance of the primary frequency control based on virtual synchronous generator (PFC-VSG) and the autonomous frequency control based on virtual synchronous generator (AFC-VSG), a grid connected system model with single VSG is built in MATLAB / Simulink environment. The regulation performance of VSG is simulated and analyzed under the sudden load increase and sudden load decrease. In this model, the DC side is 750V medium voltage DC bus, the AC side is 220V, 50Hz system, the AC load active power is 10kW, and the reactive power is 100var. The other simulation parameters are shown in Appendix Table A1.
After modification
Page 10 Line 297-304
To verify the regulation performance of primary frequency control (PFC) and autonomous frequency control (AFC) in the microgrid with single converter, the microgrid model with single converter is established in Matlab / Simulink environment. In this model, the DC side is 750V medium voltage DC bus, the AC side is 220 V, 50 Hz system, the AC load active power is 10kW, the reactive power is 100var, other simulation parameters are shown in table A1. PFC and AFC are used in the power frequency control of VSG algorithm to form two converter control strategies respectively, hence the birth of the PFC-VSG and AFC-VSG. The regulation performance of VSG is simulated and analyzed under the sudden AC load increase and sudden AC load decrease.
2.Before modification
To verify the effectiveness of the APFC-VSG control strategy when multiple VSGs operate simultaneously. A grid connected system model of three VSGs is built, and part of simulation parameters are shown in Table 4.
After modification
Page 11 Line 337-340
To verify the regulation performance among the PFC-VSG, AFC-VSG and APFC-VSG in the microgrid with multiple converters, the microgrid model with three converters is established in MATLAB / Simulink environment (as shown in Figure 3), and part of simulation parameters are shown in Table 4.
- Before modification
In Table 4, the active power droop control coefficient η is set according to the inverse ratio of the VSG damping coefficient. What’s more, the ratio of the coefficient of each proportional control λ is equal to the ratio of each VSG' s rated value. Then, a load fluctuation is added to each VSG when the simulation time is 0.1s. which makes the active power of VSG1 and VSG3 rise from 10000W to 11000W, and brings the active power of VSG2 decrease from 10000W to 8000W. The comparison of the frequency regulation performance of each control strategy is given in Figure12.
After modification
Page 11 Line 342-347
From the analysis results for the AFC-VSG in section 2, we can find that the output active power of VSG is not proportional to its rated capacity due to the secondary frequency regulation controller. In order to realize the optimal power allocation when multiple VSGs operate simultaneously, we set the active power droop control coefficient η according to the inverse ratio of the VSG damping coefficient. What’s more, the ratio of the coefficient of each proportional control λ is equal to the ratio of each VSG' s rated value.
- Before modification
Compared to the PFC-VSG control strategy, by adding the proportional control to the output of the AFC-VSG, the microgrid can realize the secondary frequency control when three VSGs operate simultaneously, and the frequency deviation is very small compared with the PFC-VSG. At the same time, in Figure 13, the proportional control in VSG allocates the system active power capacity according to the rated capacity of each VSG in microgrid. However, as shown in Figure 13, the active power oscillates, which is caused by the different damping provided by each VSG. Hence, the APFC-VSG proposed in part 2 is used to replace the AFC-VSG.
After modification
Page 12 Line 357-361
Compared to the PFC-VSG control strategy, by adding the proportional control to the output of the AFC-VSG, the microgrid can realize the secondary frequency control when three VSGs operate simultaneously, and the frequency deviation is very small compared with the PFC-VSG. At the same time, in Figure 13, the proportional control in VSG allocates the system active power capacity according to the rated capacity of each VSG in microgrid. However, as shown in Figure 13, the active power oscillates, which is caused by the different damping provided by each VSG. Hence, the APFC-VSG proposed in section 2 is used to replace the AFC-VSG. The APFC-VSG is constructed by adding the active droop controller to the input of the secondary frequency control controller. In order to eliminate the dynamic difference among multiple VSGs which is caused by damping coefficient, the coefficient η of the active power droop controller is selected according to the inverse ratio of the damping D value in each VSG.
Point 3: Table A1 is not referenced in the text
Responses: Thank you for your carefully reading of our manuscript, due to our carelessness, this mistake was caused. Following your review recommendations, we modified our manuscript. Table A1 are respectively referred in line 252, page 8 and line 304, page 10.
Other changes:
The construction of this paper is changed in revised paper and marked in red. And it would not influence the content of this paper.
We appreciate for Editors/Reviewers’ warm work earnestly, and hope that the correction will meet with approval.
Once again, thank you very much for your helpful comments and suggestions.

Reviewer 2 Report
The paper deals with a VSG-based autonomous power-frequency control strategy.
The authors develop a detailed mathematical model and test the control strategy on microgrids equipped with single or multiple VGSs.
In order to complete the analysis, in the case of multiple VGSs some tests and considerations about the robustness of the model should be developed; for example in the event of a VSG being out of order.
Author Response
Response to Reviewer 2 Comments
Dear Editors and Reviewers:
Thank you for your comments concerning our manuscript entitled “An Autonomous Power-Frequency Control Strategy Based on Load Virtual Synchronous Generator” (ID: processes-758919). Those comments are all valuable and helpful for revising and improving our paper, as well as the important guiding significance to our further researches. We have studied comments carefully and have made correction which we hope meet with approval. Revised portions are marked in red in the paper. The corrections in the paper and the responds to the reviewer’s comments are as follows.
Point 1: In order to complete the analysis, in the case of multiple VGSs some tests and considerations about the robustness of the model should be developed; for example in the event of a VSG being out of order.
Responses: Thank you for giving this constructive advice for us. We understand that some tests and considerations about the robustness of this model(for example in the event of a VSG being out of order)would better reveal the mechanism of microgrid operation. However, in present study, we mainly focused on suppressing the load fluctuation, and the proposed method APFC-VSG has no capacity of enhancing the robustness in the event of a VSG being out of order. We will take your suggestions in the next study by designing a VSG-based nonlinear robust adaptive controller, thus enhancing the robustness and adaptability of microgrid. Once again thank you for this constructive advice for our paper.
Other changes:
The construction of this paper is changed in revised paper and marked in red. And it would not influence the content of this paper.
We appreciate for Editors/Reviewers’ warm work earnestly, and hope that the correction will meet with approval.
Once again, thank you very much for your helpful comments and suggestions.
Reviewer 3 Report
This manuscript describes the autonomous power frequency control strategy based on VSG. The proposed idea is introduced well with a dynamic small-signal model and the dynamic stability analyses and their simulation shows the validity of the proposed idea. Nevertheless, it is recommended to revise it with referring the following comments.
- In introduction, the distinction between what authors purely suggested and what they did with reference to the existing papers is not clear. You may refer the following references:
- Kun Jiang, Hongsheng Su, Ding Sun, “Secondary frequency regulation scheme based on improved virtual synchronous generator in an islanded microgrid,” J. of Vibroengineering, Vol. 21, Issue 1, 2019, p. 215-227. https://doi.org/10.21595/jve.2018.19662
- Kai Shi, Guanglei Zhou, Peifeng Xu, Haihan Ye and Fei Tan, “The Integrated Switching Control Strategy for Grid-Connected and Islanding Operation of Micro-Grid Inverters Based on a Virtual Synchronous Generator,” Energies, 2018, 11, 1544; doi:10.3390/en11061544
- Yan Xu, Tengfei Zhang, Dong Yue, “Frequent Deviation-Free Control for Micro-Grid Operation Modes Switching Based on Virtual Synchronous Generator,” LSMS/ICSEE (3) 2017: 597-606
- Regarding the current research trend of this topic, authors' point of view and the publication year of references do not match. For example, in page 2, line 58 to 63, <To conquer this issue, ~ in [13], ~fills the blank in [11, 12], ~ modulation.>, but [13] was published later than [11, 12].
- From page 2, line 43 to page 3, line 46: Please describe the characteristic differences of a voltage controlled VSG and a current controlled VSG for a grid connected microgrid and islanded microgrid a little bit detail. And through this paragraph, this reviewer was misled to expect that authors would conduct the research of VSG application to a islanded microgrid.
- Please double check Eqs. (1) to (3) and Fig. 1 for correctness.
- All controller gain parameters are shown in Table A1, no mention about the design of them in this manuscript.
- Define i, j, k in Eq. (29). In page 8, line 259, there are only mentioning of 11 state variables of j.
- In page 9, line 284, how authors can say that <The dominant eigenvalue of area a is 1, 2, 10, and 11 respectively.>.
- To calculate Pij in Table 3, where are α and β values corresponding shown?
- It is recommended to show any experimental results as usual in most references.
- In Fig. 13, no figures of results of the PFC-VSG.
- Minor corrections
- The use of uppercase and lowercase letters should be unified. For example
- Page 2, line 69, what's more, ~ --> What's more
- In Table 1, stable --> Stable, and Ud and Uq --> ud and uq
- Title of Table 2, eigenvalues --> Eigenvalues
- Title of Table 3, participation --> Participation
- Caption of Fig. 9, Sudden --> sudden
- Title of Table 4, part --> Part
- Caption of Fig. 11, Sudden --> sudden
- Title of Table A1, system --> System
- In Table A1, all gain paprmeters have been used in capital in Fig. 6 and in previous equations.
- In References: Too many!
- Page 3, line 113 to 115, <After the introduction, a VSG-based autonomous power-frequency control strategy is proposed in Section 2. Then, a dynamic small-signal model of the microgrid with single VSG is established in Section 3.>
- Page 3, line 124 to 125, Correct <θ and D respectively signifies purchased the actual
power angle of the system and the damping coefficient> - Page 3, line 125, Correct <What ‘more,> --> What's more,
- Correct references.
- The use of uppercase and lowercase letters should be unified. For example
Author Response
Response to Reviewer 3 Comments
Dear Editors and Reviewers:
Thank you for your letter and for the reviews’ comments concerning our manuscript entitled “An Autonomous Power-Frequency Control Strategy Based on Load Virtual Synchronous Generator” (ID: processes-758919). Those comments are all valuable and helpful for revising and improving our paper, as well as the important guiding significance to our researches. We have studied comments carefully and have made correction which we hope meet with approval. Revised portions are marked in red in the paper. The corrections in the paper and the responds to the reviewer’s comments are as follows.
Point 1: In introduction, the distinction between what authors purely suggested and what they did with reference to the existing papers is not clear. You may refer the following references:
A.Kun Jiang, Hongsheng Su, Ding Sun, “Secondary frequency regulation scheme based on improved virtual synchronous generator in an islanded microgrid,” J. of Vibroengineering, Vol. 21, Issue 1, 2019, p. 215-227. https://doi.org/10.21595/jve.2018.19662
B.Kai Shi, Guanglei Zhou, Peifeng Xu, Haihan Ye and Fei Tan, “The Integrated Switching Control Strategy for Grid-Connected and Islanding Operation of Micro-Grid Inverters Based on a Virtual Synchronous Generator,” Energies, 2018, 11, 1544; doi:10.3390/en11061544
C.Yan Xu, Tengfei Zhang, Dong Yue, “Frequent Deviation-Free Control for Micro-Grid Operation Modes Switching Based on Virtual Synchronous Generator,” LSMS/ICSEE (3) 2017: 597-606Authors should improve the state of art. There are more related works.
Responses: Thank you for giving this constructive advice for us, we have made the following changes based on the comments.
page 2 Line 55-63
To enhance the power quality of the microgrid, many researchers put forward a series of improved algorithm of VSG to realize the secondary frequency regulation. Reference [12] introduces a PI controller to the power-frequency control unit in the VSG, so that the system can automatically realize the secondary frequency regulation in island mode. In [13], by bringing two integrators in the virtual mechanical and damping torque part in the p-f controller of the traditional VSG, the system frequency can return to the rated value in an islanded microgrid. In addition, reference [14] proposes an integrated control algorithm, which combines a quasi-synchronization algorithm with an islanding detection algorithm, thus achieving the frequent deviation-free regulation in both grid-connected microgrid and islanded microgrid.
- Xu Y.; Zhang T. F.; Yue D. Frequent Deviation-Free Control for Micro-Grid Operation Modes Switching Based on Virtual Synchronous Generator. LSMS/ICSEE, Nanjing, Jiangsu, China, 22-24 Sept. 2017.
- Jiang K.; Su H. S.; Sun D. Secondary Frequency Regulation Scheme Based on Improved Virtual Synchronous Generator in an Islanded Microgrid. Journal of Vibroengineering 2019, 21, 215-227.
- Shi K.; Zhou G. L.; Xu P. F.; Ye H. H.; Tan F. The Integrated Switching Control Strategy for Grid-Connected and Islanding Operation of Micro-Grid Inverters Based on a Virtual Synchronous Generator. Energies 2018, 11, 1544.
Point 2: Regarding the current research trend of this topic, authors' point of view and the publication year of references do not match. For example, in page 2, line 58 to 63, <To conquer this issue, ~ in [13], ~fills the blank in [11, 12], ~ modulation.>, but [13] was published later than [11, 12].
Responses: Thank you for giving this constructive advice for us, due to our carelessness, this mistake was caused. So, we have made changes as follows.
page 1 Line 42-52
Many ameliorative strategies have been proposed to enhance the microgrid stability. In [9], a VSG-based auto disturbance rejection control strategy is proposed. Simulation experiments verify the positive effects of the proposed control strategy in estimating and compensating the total disturbance of the system. Moreover, reference [10] presents an adaptive virtual inertia control strategy based on improved bang-bang control. On the one hand, it can make full use of the variability of the virtual inertia to reduce dynamic frequency deviation. On the other hand, the frequency stability of the system is improved by setting the steady-state frequency interval and the steady-state inertia. In [11], a self-adaptive inertia and damping combination control strategy is presented, which supports the short-term active power of the system by using the adaptive inertial damping, and realizes the adaptive inertial damping control of the microgrid primary frequency modulation.
Point 3: From page 2, line 43 to page 3, line 46: Please describe the characteristic differences of a voltage-controlled VSG and a current-controlled VSG for a grid connected microgrid and islanded microgrid a little bit detail. And through this paragraph, this reviewer was misled to expect that authors would conduct the research of VSG application to a islanded microgrid.
Responses: Thank you for giving this constructive advice for us, after careful examination, we think there are some logical problems in this paragraph. In addition, in order to highlight the research focus of this paper, we directly introduce the concept of VSG to avoid misleading readers. we replace the original paragraph as follows.
Before modification
The existing modified control strategies of the grid-connected converters mainly include droop control and VSG control strategy [7]. However, in the islanded microgrid, it is difficult for the droop-based converters to provide inertia for the microgrid, which is more sensitive to the load fluctuation and makes the frequency change dramatically [8]. In contrast, the VSG-based converters have the capacity of changing the frequency smoothly in the same situation. Therefore, the VSG control strategy can satisfy the dynamic characteristics of the islanded microgrids. Moreover, VSG technology is mainly composed of the voltage-controlled VSG and current-controlled VSG [9]. In the grid-connected microgrid, both two strategies can effectively restrain the load fluctuations in the power grid. However, in the islanded microgrid, a current-controlled VSG only equates VSG with a current source, which cannot provide complete voltage and frequency support for the microgrid. On the contrary, the voltage-controlled VSG is designed according to the rotor motion equation and reactive voltage regulation control characteristics, which can better simulate the external characteristics of synchronous generators [10].
After modification
page 1 Line 37-43
For instance, Prof. Zhong puts forward the concept of VSG [5-8]. In the context of backbone power grid being the AC power grid, synchronous generators are defined as the default mechanism of the system, so all players in this system have autonomous synchronous characteristics. Adding a second-order model of SGs into the control strategy of the power converter, hence the birth of the virtual synchronous generator (VSG), would be helpful in making power converters more efficient and friendlier to AC power grid. Based on Prof. Zhong's researches, many ameliorative strategies have been proposed to enhance the microgrid stability.
Point 4: Please double check Eqs. (1) to (3) and Fig. 1 for correctness.
Responses: Thank you for your careful reading of our manuscript, due to our carelessness, this mistake was caused. We have checked the Eqs. (1) to (3) and Fig. 1, and the modifications are as follows.
|
(1) |
|
|
(2) |
|
|
(3) |
|
|
(4) |
Point 5: All controller gain parameters are shown in Table A1, no mention about the design of them in this manuscript.
Responses: Thank you for giving this constructive advice for us. We are sorry for that controller gain parameters are designed by experiment without calculation, In the next research, we will follow your comment to design the parameters by calculating. In addition, we have supplemented the design process of controller gain parameters in the appendix as follows.
page 13 Line 398-415
All controller gains in Table A1 are designed as follows
- Kf,
- The proportional coefficient Kf increases from 0 until the system oscillates. Then, from the Kf at this time gradually decreases until the system oscillation disappears, record the value of Kf when the system oscillation disappears, and set the proportional coefficient Kf of the proportional controller to 60% - 70% of the Kf Value.
- kp1, kp2
When the parameter Kp is determined in the PI controller, the integral term is first removed to make it a simple proportional regulation. The input is set to 60% - 70% of the maximum allowable output value, and the proportional coefficient Kp increases from 0 until the system oscillates. Then, from Kp at this time gradually decreases until the system oscillation disappears, record the value of Kp when the system oscillation disappears, and set the proportional coefficient of the PI controller to 60% - 70% of the Kp Value.
- Ki, Ki1, Ki2.
After the proportional coefficient Kp is determined, set a large integral constant Ti, in which Kp / Ti = Ki, then gradually reduce Ti until the system oscillates, and then gradually increase Ti in turn until the system oscillation disappears, record the Ti value when the oscillation disappears, and set Ti of the PI controller to 150% - 180% of this Ti value , thus obtaining the Ki.
Point 6: Define i, j, k in Eq. (29). In page 8, line 259, there are only mentioning of 11 state variables of j.
Responses: Thank you for your careful reading of our manuscript, due to our carelessness, this mistake was caused. We have defined the i, j, k as follows.
page 9 Line 276-278
Where j and i represent the state variable and the response mode respectively, and N is the number of the eigenvalues. In addition, αkj represents the left eigenvector of the j-th state variable in the k-th mode, and β kj represents the right eigenvector of the j-th state variable in the k-th mode.
Point 7: In page 9, line 284, how authors can say that <The dominant eigenvalue of area a is 1, 2, 10, and 11 respectively.>.
Responses: Thank you for your careful reading of our manuscript, due to our carelessness, this mistake was caused. We have changed this sentence as follows.
page 9 Line 280-281
From the distribution of the eigenvalues in Figure 7, the eigenvalue of area a is 1, 2, 10 and 11 respectively.
Point 8: To calculate Pij in Table 3, where are α and β values corresponding shown? It is recommended to show any experimental results as usual in most references.
Responses: Thank you for giving this constructive advice for us. We are so sorry that the calculated values of α and β are not saved because they are used as intermediate variables in the program, and the original program has been rewritten, we will learn a lesson and pay attention to storing the original data in the next research.
Point 9: In Fig. 13, no figures of results of the PFC-VSG.
Responses: Thank you for your careful reading of our manuscript, due to our carelessness, this mistake was caused. We have changed this Fig.13 as follows.
Point 10: Minor corrections
- The use of uppercase and lowercase letters should be unified. For example
- Page 2, line 69, what's more, ~ --> What's more
- In Table 1, stable --> Stable, and Ud and Uq --> ud and uq
- Title of Table 2, eigenvalues --> Eigenvalues
- Title of Table 3, participation --> Participation
- Caption of Fig. 9, Sudden --> sudden
- Title of Table 4, part --> Part
- Caption of Fig. 11, Sudden --> sudden
- Title of Table A1, system --> System
- In Table A1, all gain paprmeters have been used in capital in Fig. 6 and in previous equations.
- In References: Too many! C. Page 3, line 124 to 125, Correct <θ and D respectively signifies purchased the actualE. Correct references.A.
- 1.page 2 Line 78
- Responses: Thank you for your careful reading of our manuscript, due to our carelessness, these mistakes were caused. We have modified these mistakes as follows.
- D. Page 3, line 125, Correct <What ‘more,> --> What's more,
- B. Page3, line 113 to 115, <After the introduction, a VSG-based autonomous power-frequency control strategy is proposed in Section 2. Then, a dynamic small-signal model of the microgrid with single VSG is established in Section 3.>
what's more, ~ --> What's more
2.In Table 1
stable --> Stable, and Ud and Uq --> ud and uq
3.In Table 2
Title of Table 2, eigenvalues --> Eigenvalues
4.In Table 3
Title of Table 3, participation --> Participation
5.In Caption of Fig. 9
Sudden --> sudden
- Title of Table 4
part --> Part
- Caption of Fig. 11
Sudden --> sudden
- Title of Table A1
system --> System
9.In Table A1
All gain parameters have been changed into uppercase in Table A1.
10.In References
All references have been revised.
B.
page 2 Line 104-105
The remaining of this paper is organized as follows. After the introduction, a VSG-based autonomous power-frequency control strategy is proposed in Section 2. Then, a dynamic small-signal model of the microgrid with single VSG is established in Section 3. In Section 4, the dynamic stability of the model is analyzed, and then the simulation processes are given in Section 5. Finally, some brief conclusions are listed.
C.
page 2 Line 114-115
J represents the moment of inertia of the system. θ and D represent the actual power angle of the system and the damping coefficient respectively.
D.
page 2 Line 115
<What ‘more,> --> What's more,
E.
All errors in references have been corrected.
Special thanks to you for your good comments.
Other changes:
The construction of this paper is changed in revised paper and marked in red. And it would not influence the content of this paper.
We appreciate for Editors/Reviewers’ warm work earnestly, and hope that the correction will meet with approval.
Once again, thank you very much for your helpful comments and suggestions.

Reviewer 4 Report
- English: Overall your English is good.
- In page 3 line 125: change “What’ more” with “What’s more”.
- In page 3 line 130: change “Kfis” with “Kf is”.
- In page 3 line 133: change “according to the equation (1) (2)” with “according to the equation (1) and (2)”.
- In page 3 Equation (3): please explain equations separately.
and not
Please add “with n and T are” before expressions of n and T respectively.
- In page 4 line 144: change “Kfand” with “Kf and”.
- In page 4: from equation (5) correct numbering of all equations: equation (5) becomes (4) and so on…
- In page 6 Line 212 : in the expression of P there is no sign (-) please verify.
- In page 8 Line 256 and 257: please rectify “The AC load capacity is 10kW, and the reactive capacity is 100var” with “the AC load active power is 10kW, and the reactive power is 100var”.
- In page 9 lines from 275 to 277: please restructure your phrase: use the active voice.
- In page 9 lines from 277 to 278: “we can find that the influence of each local control link of the system on the stability of the system.” Incomplete phrase.
- In page 10 line 306: change “0.2 KW” to “2 KW”.
- In page 10 line 308: change “1.05 KW” to “10.05 KW”.
- In page 10 line 310: change “1 KW to 1.2 KW” to “10 KW to 12 KW”.
- In page 11 line 322: change “0.2 KW” to “2 KW”.
- In page 11 line 324: change “0.95 KW” to “9.5 KW”.
- In page 11 line 326: change “1 KW to 0.8 KW” to “10 KW to 8 KW”.
- In page 11 In table 4: please rectify “KW” with “W” as it is used in the text below the table.
- Following these remarks, it is recommended to change the unit from Watt to KW in all text, figures and tables.
- In page 12: Figure 14 is not so clear.
- In the references list please follow the Template and do not use “et al” in the references 12, 13, 14, 15, 16, 17, 18, 21 and 22.
- Please list all the authors in one reference:
for example: reference 4 misses one author “Burak Ozpineci”.
Conclusion: Overall, your paper is good enough to be published.

Author Response
Response to Reviewer 4 Comments
Dear Editors and Reviewers:
Thank you for your letter and for the reviews’ comments concerning our manuscript entitled “An Autonomous Power-Frequency Control Strategy Based on Load Virtual Synchronous Generator” (ID: processes-758919). Those comments are all valuable and helpful for revising and improving our paper, as well as the important guiding significance to our researches. We have studied comments carefully and have made correction which we hope meet with approval. Revised portions are marked in red in the paper. The corrections in the paper and the responds to the reviewer’s comments are as follows.
Point 1: In page 3 line 125: change “What’ more” with “What’s more”.
Responses:
page 2 Line 78
what's more, ~ --> What's more
Point 2: In page 3 line 130: change “Kfis” with “Kf is”.
Responses:
page 3 Line 119
Kfis~ --> Kf is
Point 3: In page 3 line 133: change “according to the equation (1) (2)” with “according to the equation (1) and (2)”.
Responses:
page 3 Line 122
according to the equation (1) (2) ~ --> according to the equation (1) and (2).
Point 4: In page 3 Equation (3): please explain equations separately. and not Please add “with n and T are” before expressions of n and T respectively.
|
(3) |
|
|
(4) |
Point 5: In page 4 line 144: change “Kfand” with “Kf and”.
Responses:
page 4 Line 136
Kfand~ --> Kf and
Point 6: In page 4: from equation (5) correct numbering of all equations: equation (5) becomes (4) and so on…
Responses: The number of all equations are corrected in this paper.
Point 7: In page 6 Line 212 : in the expression of P there is no sign (-) please verify.
Responses: It is verified that there is no sign (-) in the expression of P.
Point 8: In page 8 Line 256 and 257: please rectify “The AC load capacity is 10kW, and the reactive capacity is 100var” with “the AC load active power is 10kW, and the reactive power is 100var”.
Responses:
page 8 Line 249-250
The AC load capacity is 10kW, and the reactive capacity is 100var ~ --> the AC load active power is 10kW, and the reactive power is 100var
Point 9: In page 9 lines from 275 to 277: please restructure your phrase: use the active voice.
Responses:
page 9 Line 268-270
Pij is the participation factor. As Pij gradually increases, it indicates that the influence of the state variable j on the response mode i corresponding to the characteristic value gradually increases.
Point 10: In page 9 lines from 277 to 278: “we can find that the influence of each local control link of the system on the stability of the system.” Incomplete phrase.
Responses:
page 9 Line 270-271
By judging the influence degree of different state variables on eigenvalues, we can find the influence of each sub model on the stability of the system.
Point 11: In page 10 line 306: change “0.2 KW” to “2 KW”.
Responses:
page 10 Line 309
0.2 kW ~ --> 2 kW.
Point 12: In page 10 line 308: change “1.05 KW” to “10.05 KW”.
Responses:
page 10 Line 311
1.05 kW ~ --> 10.5 kW.
Point 13: In page 10 line 310: change “1 KW to 1.2 KW” to “10 KW to 12 KW”.
Responses:
page 10 Line 313
1 kW to 1.2 kW ~ --> 10kW to 12kW.
Point 14: In page 11 line 322: change “0.2 KW” to “2 KW”.
Responses:
page 10 Line 325
0.2 kW ~ --> 2 kW.
Point 15: In page 11 line 324: change “0.95 KW” to “9.5 KW”.
Responses:
page 10 Line 326
0.95 kW ~ --> 9.5 kW.
Point 16: In page 11 line 326: change “1 KW to 0.8 KW” to “10 KW to 8 KW”.
Responses:
page 10 Line 327
1 kW to 0.8 kW” to “10 kW to 8 kW”.
Point 17: In page 11 In table 4: please rectify “KW” with “W” as it is used in the text below the table.
Responses:
Table 4
We have changed the unit of active power in Table 4 as follows.
|
Rated value of the active power (kW) |
|
10 |
|
6 |
|
10 |
Point 18: Following these remarks, it is recommended to change the unit from Watt to KW in all text, figures and tables.
Responses:
We have changed the unit from Watt to KW in all text, figures and tables.
Point 19: In page 12: Figure 14 is not so clear.
Responses:
We modified Figure 14 as follows.
Before modification After modification
Point 20 and 21. In the references list please follow the Template and do not use “et al” in the references 12, 13, 14, 15, 16, 17, 18, 21 and 22. Please list all the authors in one reference:
for example: reference 4 misses one author “Burak Ozpineci”.
Responses to 20 and 21: Thank you for giving this constructive advice for us, due to our carelessness, this mistake was caused. All the formats of references have been modified in this paper.
Special thanks to you for your good comments.
Other changes:
The construction of this paper is changed in revised paper and marked in red. And it didn’t influence the content of this paper.
We appreciate for Editors/Reviewers’ warm work earnestly, and hope that the correction will meet with approval.
Once again, thank you very much for your comments and suggestions.

Round 2
Reviewer 2 Report
Dear authors, the proposed development of the study concerns the design of an adaptive controller. This target is highly appreciated and can represent an important milestone. I wish you all the best in your work.Author Response
Thank you for your high recognition of our next research. we wish you all the best in your work.
Reviewer 3 Report
Thank you for revision of this manuscript. Nevertheless there are something left to make clear. Please refer the followings:
- From Eqs. (1) and (2), how can you get the control block diagram of Fig. 1? It doesn't match. Let's show the derivation process.
- Regarding Eqs. (3) and (4):
Refer the attached file, please.

Author Response
Thank you again for your comments and the responds are in the file(responds to reviewer (ID processes-758919.pdf))

Round 3
Reviewer 3 Report
Revised well.